# A Hepatic Scaffold from Decellularized Liver Tissue: Food for Thought

**DOI:** 10.3390/biom9120813

**Published:** 2019-12-02

**Authors:** Stefania Croce, Andrea Peloso, Tamara Zoro, Maria Antonietta Avanzini, Lorenzo Cobianchi

**Affiliations:** 1Department of Clinical, Surgical, Diagnostic & Pediatric Sciences, University of Pavia, 27100 Pavia, Italy; stefania_croce186@yahoo.it (S.C.); tamara.zoro01@universitadipavia.it (T.Z.); lorenzo.cobianchi@unipv.it (L.C.); 2Fondazione IRCCS Policlinico S. Matteo, Immunology & Transplantation Laboratory/Pediatric Oncohematology/Cell Factory, 27100 Pavia, Italy; ma.avanzini@smatteo.pv.it; 3Hepatology and Transplantation Laboratory, Department of Surgery, Faculty of Medicine, University of Geneva, 1205 Geneva, Switzerland; 4Divisions of Abdominal and Transplantation Surgery, Department of Surgery, Geneva University Hospitals, 1205 Geneva, Switzerland; 5General Surgery Department, Fondazione IRCCS Policlinico San Matteo, 27100 Pavia, Italy

**Keywords:** scaffold, regenerative medicine, liver, extracellular matrix, liver bioengineering

## Abstract

Allogeneic liver transplantation is still deemed the gold standard solution for end-stage organ failure; however, donor organ shortages have led to extended waiting lists for organ transplants. In order to overcome the lack of donors, the development of new therapeutic options is mandatory. In the last several years, organ bioengineering has been extensively explored to provide transplantable tissues or whole organs with the final goal of creating a three-dimensional growth microenvironment mimicking the native structure. It has been frequently reported that an extracellular matrix-based scaffold offers a structural support and important biological molecules that could help cellular proliferation during the recellularization process. The aim of the present review is to underline the recent developments in cell-on-scaffold technology for liver bioengineering, taking into account: (1) biological and synthetic scaffolds; (2) animal and human tissue decellularization; (3) scaffold recellularization; (4) 3D bioprinting; and (5) organoid technology. Future possible clinical applications in regenerative medicine for liver tissue engineering and for drug testing were underlined and dissected.

## 1. Introduction

Liver dysfunction is one of the most severe health problems worldwide and is characterized by high morbidity and mortality [1]. Allogeneic liver transplantation is still considered the gold standard solution for end-stage organ failure, such as end-stage liver disease, providing a better quality of life in addition to cost effectiveness; however, a shortage of donor organs has resulted in extended transplantation waiting lists. In detail, data from the United States show that more than 15,000 patients have been added to the waiting list for liver transplantation, while only around 6000 liver transplants are actually performed yearly. This results in an increasing mismatch between the numbers of liver donors and recipients [2,3,4]. In order to overcome the lack of donors, the development of new therapeutic options, such as cell-based therapies that include liver cell transplantations, bioartificial livers, and engineered hepatic tissues, is mandatory [5,6,7]. Hepatocyte transplantation has been tested for more than 20 years; one approach involved injections of primary hepatocytes into the hepatic parenchyma with the aim of restoring liver activity and metabolic functions. Although the replacement of 5% of liver cells with primary hepatocytes could significantly improve liver function, the general problem with this approach is the limited repopulation capacity of engrafted cells partly due to instant blood-mediated inflammatory reaction (IBMIR) [8]. Moreover, the inability of healthcare professionals to monitor graft health and the frequent cases of rejection limit the use of hepatocyte transplantation in clinical practice [6]. A liver regenerative medicine-based approach (bioartificial devices (BALs) and a liver on-a-chip platform) has been recently explored, aiming to temporarily support the organ until transplantation. BALs could eliminate toxins accumulating in liver failure and supply liver cells supporting the organ with respect to synthetic and regulatory functions [9]. An organ-on-a-chip is a microfluidic cell culture device that allows the seeded cells to simulate the multicellular microenvironments and vascular perfusion of the body, which is not possible with conventional 2D or 3D culture systems. Moreover, this technology may have a great potential in the study of tissue/organ physio-pathology and drug development as an alternative to animal testing. Until now, in the context of liver dysfunction, the liver-on-chip platform has been applied in a murine model, obtaining an increase in graft function as indicated by the production of multiple human liver proteins and the emergence of characteristic liver stereotypical microstructures, which have been histologically demonstrated [10]. This device was able to mimic physiological fluid flow conditions and control temporal and spatial distribution of nutrients and growth factors to cells in a model of chronic liver disease [11,12]. However, both BAL and the organ-on-a-chip may not represent a long-term substitute for liver transplantation. In order to overcome this limitation, organ bioengineering (OBE) has been developed in order to provide an inexhaustible supply of transplantable tissues or whole organs. OBE includes all the strategies that allow the recapitulation of macromolecules and vascular system structure starting from a synthetic or biological scaffold. The least common denominator of this procedure assumes that a variety of cells can be seeded onto a suitable scaffold in a specific growth and differentiation environment. These scaffolds result from synthetic or biological sources. Synthetic scaffolds are produced using different techniques including electrospinning [13], three-dimensional (3D) bioprinting technology [14], and hydrogel-based technologies [15]. Most of these technologies are also not used directly for transplantation purposes but rather to overcome the drawbacks of in vitro liver testing during drug development. On the other side, biological scaffolds are obtained from discarded organs through different decellularization methods [16,17]. Regenerative medicine (RM) has shown the hypothetical potential to overcome the limit of organ availability and to allow transplant without immunosuppression. The bioengineering of whole organs is an exciting area of transplantation research. There have been considerable advances that have led to the development of a suitable decellularized scaffold in small animal, porcine, and human models. The aim of the present review is to inspect the main strategies in the field of OBE, focusing on the options for liver functional reconstitution. In particular, we consider the current progress in scaffold-base technology taking into account: (1) biological and synthetic scaffolds; (2) animal and human tissue decellularization; (3) scaffold recellularization; (4) 3D bioprinting; and (5) organoid technology. Future possible clinical applications in regenerative medicine for liver tissue engineering and for drug testing have been underlined and dissected (Figure 1).

## 2. Method

A literature search from the databases PubMed, Google Scholar, and Science Direct was done up to March 2019, although we principally focused on the later years to provide the latest technological advances for several key-words inquiries: (1) liver; (2) scaffold; (3) extracellular matrix; (4) biomaterials; (5) 3D printing; (6) tissue; (7) organs; (8) regenerative medicine; (9) stem cells; (10) tissue engineering; (11) liver transplantation; (12) 3D liver bio-printing; and (13) animal models. These databases mostly use English as the main language. Duplicate articles were removed, and only full articles containing some of the above-mentioned words were included.

## 3. Regenerative Medicine and Cell-On-Scaffold Technology

RM and OBE share the same goal of trying to replace human tissues or organs towards the re-establishment of its native function [18,19]. The RM paradigm consists of three important factors: (1) a supporting 3D scaffold; (2) cells (parenchymal and vascular); and (3) signaling molecules. Specifically, the scaffold offers a structural, biochemical, and biomechanical architecture to guide and regulate cell behavior and tissue development together with integrated signaling organ-specific molecules. As previously reported, liver transplantation is still considered the only successful treatment both for chronic end-stage liver disease in addition to acute liver failure [2,3]. Shortage of organs still remains an unsolved problem leading to use marginal grafts as an option to increase the organ supply [4].

RM/OBE tries to overcome this limitation by producing bioengineered organs that are capable of supporting hepatic physiological functions, including detoxification, protein synthesis, and the production of bile, which is necessary for digestion. As a corollary, this approach aims to complete the so-called “halfway technology”. Halfway technology was introduced by Lewis Thomas and refers to a treatment that “represents the kinds of things that must be done after the fact, in efforts to compensate for the lack of understanding of the mechanisms involved in a disease process” [20]. In other words, halfway technology refers to treatments that treat symptoms without really curing the underlying disease. In transplantation jargon, this can be applied to liver transplantation when performed in a patient with end-stage liver disease associated with hepatitis C virus (HCV) [21] (before the introduction of sofosbuvir [22]) or in a patient with colorectal metastases [23]. Second, the life-long need for immunosuppression therapy may potentially lead to severe acute or chronic toxicity, causing additional clinical syndromes [24]. For these reasons the potential application of RM/OBE as an endless source of patient-specific organs (and consequently an immunosuppression-free state) could be ground-breaking.

Cell-on-scaffold technology is the cornerstone of RM/OBE and is obtained by the decellularization technique pioneered by Ott et al. in 2008 [25], who perfused a rat heart with specific chemical detergents (sodium dodecyl sulphate and Triton X-100) to create a decellularized “ghost” heart composed of just extracellular matrix (ECM). This ECM-based scaffold provides not only a structural support to the native anatomy but also supplies important biological molecules that could support cellular proliferation during the recellularization process. These results paved the way for the application of this technology to other organs, producing biological, bio-active, and three-dimensional organ-specific scaffolds. Whole-organ detergent-perfusion protocols permit obtaining an acellular natural three-dimensional framework by removing all the cells from several organ such as the liver, kidney, and pancreas [17,26]. In this way, it is possible to create an ideal surface for cellular growth. In fact, these scaffolds are composed of organ-specific ECM that contain growth factors important for cellular proliferation and function [27,28]. Moreover, vascular access to native organs is preserved and may be incorporated in the recipient’s system after the implant.

As the aim of this review, advantages and limits of this technique as applied to liver bioengineering will be discussed in the following sections.

### 3.1. The Importance of the Third Dimension for Cell Growth and Proliferation

Bi-dimensional cellular cultures have been widely used worldwide for decades. March et al. [29] described a stable, robust, and reliable in vitro primary human hepatocyte model for infectious disease applications. Using this system, they reproduced in vitro liver infections, showing host–pathogen interactions, useful for drug screening and vaccine development.

However, these types of cultures are considered not appropriate in organ transplantation since the physiological architecture of a tissue is not reproduced. Inserting a third dimension into a cell culture became relevant to address this challenge and closely represent in vivo physiological settings. On the flip side, it requires a more complex multidisciplinary approach and multidisciplinary expertise to design an appropriate scaffold to support cellular organization. The final goal is to create a 3D growth microenvironment mimicking native tissue as closely as possible. At the same time, the scaffold should be porous enough to permit oxygen and nutrient delivery to seeded cells and to guarantee a physiological outlet of waste cell-derived metabolites [30].

### 3.2. Synthetic Versus Biological Scaffold

It has been well established that cells adapt to their surrounding microenvironment through local signals via the activation (or the suppression) of specific pathways concerning cellular proliferation, differentiation, and function [31]. In the literature, different studies showed the way in which 3D culture conditions could enhance the capability of cell growth [32]. As of now, a comparison between synthetic or biological scaffold has led controversial results with advantages and disadvantages in both systems [33] (Table 1).

In recent years, the development of biomaterial production technologies has improved the characteristics of synthetic scaffolds, making them suitable for repopulation process in OBE. They can be easily sterilized before clinical applications to avoid infections, are highly economical, and they are easy to synthesize. In addition, the use of synthetic scaffolds does not require organ donors [34]. A 3D structure can be fabricated starting from diversified biomaterial in order to reproduce physical and chemical proprieties of the native ECM that represents the physiological, native cellular environment. A required characteristic of 3D culture with synthetics biomaterials is to provide an environment that facilitates the nutrient and soluble factor diffusion for good cell growth [35] since the absence of a dedicated vascular tree may direct reseeded cells to apoptosis. However, it remains difficult to reproduce the complexity and the dynamic characteristics of targeted organs. Biocompatibility is mandatory for any synthetic scaffolds, in order to avoid host immune reaction. This requirement could be obtained by using ECM-derived scaffolds obtained through the organ decellularization. They provide the same environment as the native organ, including blood and lymphatic vessel structures [36] and active molecules, such as peptides and ECM-specific proteins useful for cell growth, that are difficult to reproduce artificially [37]. In fact, it has been reported that ECM bioscaffolds retain specific growth factors, cytokines, and/or chemokines that facilitate cell attachment, tissue integration, remodeling, and differentiation.

### 3.3. Synthetic Scaffold for Liver Bioengineering

Synthetic scaffolds play an important role in liver engineering. An ideal synthetic scaffold must be a bioactive substrate capable of reproducing biophysical and biochemical characteristics of liver ECM. As reported above, synthetic scaffolds should provide a structural support function, promote cell viability and proliferation, and recreate an environment suitable for the diffusion of oxygen, nutrients, and cell growth factors [38]. Liver synthetic scaffolds may be manufactured from several types of biomaterials. The material must be biocompatible, biodegradable, non-toxic, and it must not generate adverse reactions once implanted in the body. Moreover, using these synthetic sources, no pathogenicity due to animal-derived materials would arise and good reproducibility is guaranteed in large-scale production [39]. Different biomaterials, such as polyethylene glycol (PEG), poly-l-lactic acid (PLLA), polycaprolactone (PCL), and thermoplastic polyurethane (PU), have been used for creating synthetic scaffolds in hepatic bioengineering [15,40,41,42]. Moreover, the choice of the technique used is another important element to be evaluated for synthetic scaffold production. In liver tissue engineering when exploiting synthetic scaffolds, the most commonly used methods are hydrogel-based technology [43], electrospinning [44], nanofibers [13], and 3D-bioprinting [45]. Hydrogels are biocompatible materials that have generated an increasing interest over the last several decades. They are characterized by softness and possibility to be conjugated with proteins, such as collagen(s) and elastin that could enhance cell proliferation and biomolecule delivery [30,35]. These properties make them more similar to ECM than synthetic biomaterials reported above. [30,46] In a recent study, Ying Luo et al. [35] used hydrogel nanofibers to address spheroid-induced pluripotent stem cells (iPSCs) to differentiate into hepatocyte-like cells (HLCs). Their results suggest that the hydrogel culture system favored the development of aggregated iPSCs in spheroids. After 11 days of culture, human iPSCs produce spheroids (d = 50–70 µm) with a viability of 97.5%. The results showed that hydrogels also promote HLC differentiation more efficiently compared to a 2D culture system. In fact, the secretion of albumin (ALB), urea production, glycogen synthesis, and cytochrome P450 (CYP450) activity were significantly higher than under 2D conditions. Three-dimensional nanofiber scaffolds formed by electrospinning may represent a promising option for liver tissue engineering. Nanofibers create an artificial network that mimic ECM. A variety of biodegradable synthetic polymers have been used for scaffold production. PLLA is a biocompatible polymer widely used in tissue reconstruction because of its biodegradability, mechanical properties, and non-toxic nature [47].

### 3.4. Biological Scaffold for Liver

The use of ECM-based scaffolds is becoming more and more attractive in RM and organ engineering strategies due to whole organ decellularization [25,48]. The decellularization approach consists of the complete removal of cells from tissues or organs [16]. This procedure generates an ECM-based, acellular, 3D scaffold that maintains intact native organ-specific structures in terms of both hierarchical anatomical geometry and bioactive macro- and micro-molecules. ECM is the secreted product of each organ-specific resident cell, representing the ideal scaffold for the repopulation step. Composed of extracellular macromolecules such as collagen(s), elastin, laminin, glycosaminoglycans, and fibronectin in different concentrations, the ECM provides structural and biochemical support of the surrounding cells. Cell–cell communication, cell–matrix adhesion, new ECM formation, and site-appropriate differentiation of progenitor cells are mediated by ECM [49,50,51]. Consequently ECM-derived scaffolds provide cues that influence cell migration, proliferation, and differentiation [16]. This hypothesis has been confirmed by a study authored by Uygun et al. [52], in which human liver ECM-scaffolds were shown to retain matrix-bound growth factors, such as vascular endothelial growth factor (VEGF), basic fibroblast growth factor (bFGF), hepatocyte growth factor (HGF), and epidermal growth factor (EFG) that play a pivotal role in hepatocyte differentiation and function [53].

## 4. Decellularization Technology

According to the different characteristics of tissues and organs, several protocols have been reported for decellularization based on the use of different agents. In particular, decellularization could be performed through physical, chemical, and biological agents. Regardless of the decellularization technique, a good balance between cellular removal and the preservation of matrix quality is important; excessive decellularization could damage the ECM matrix thus causing biomolecule denaturation and/or the micro-architectural degradation [26]. Therefore, a good combination of cell–detergent contact time other than detergent concentration should be refined in order to optimize decellularization and reduce undesirable effects [48]. To date, perfusion decellularization seems to represent the best option for obtaining whole-liver scaffolds. This approach exploits the use of the native vascular tree sensing as the best road to disperse the detergent homogenously inside the tissue or organ. Theoretically, this method likewise allows the same contact time between cells and detergent in all the portions of the perfused organ. Perfusion-based decellularization has the big advantage of producing a whole organ scaffold with a size according to organ source.

### 4.1. Decellularization Technology Applied to Liver Engineering

Thanks to this approach, we are now able to produce small/large animal size liver scaffold [54,55] in addition to a human whole-hepatic scaffold [56]. The possibility to manipulate a whole organ, such as the liver or a lobe derived from it, is essential for transplantation purposes. Different studies indicate that perfusion-based whole-organ decellularization can retain both the native configuration and the macroscopic 3D architecture of the liver, guaranteeing biocompatibility for complete recellularization [57]. Preclinical animal models have been established in order to evaluate the efficacy of liver bioengineering starting with small animal models and progressing to human-scale hepatic scaffolds (Figure 2).

#### 4.1.1. Small Animal Models

Rat liver decellularization was performed for the first time by Uygun et al. [52] in 2010. Through a single-detergent based, portal vein antegrade perfusion, the authors were able to achieve a whole-liver, acellular ECM-based scaffold. This technology was then applied by Shupe et al. [58] who performed a similar type of experiment obtaining the same results. In the first study, rat liver was perfused with 0.1% SDS that macroscopically led to a translucent, whitish liver. Histology showed the absence of DNA with the retention of microvasculature, ECM ultrastructure, and constituents, such as collagens I and IV, fibronectin, and laminin. Shupe et al. preferred to use increasing concentrations of Triton X-100 followed by 0.1% SDS and serum in a perfusion-based protocol. Similar to the Uygun study, the investigation by Shupe et al. showed an absence of cellular compartment by hematoxylin and eosin (H&E) staining and retention of collagen IV and laminin within the ECM. Additionally, Uygun et al. demonstrated the preservation of hepatocyte functions, such as synthesis of lactate dehydrogenase (LDH) and albumin and production of urea persisting up to 8 h after heterotopic implantation of recellularized liver. From this study, some major limitations were also highlighted, such as a perfusion flow rate too slow to transport the hepatocytes to the inner parts of liver lobes, and the relatively rapid massive intravascular thrombosis leading to the final graft loss. However, from these groundbreaking studies, many other applications have been tested both in mice [59,60,61] and rat models [62,63,64].

From the simple concept of decellularization many improvements have been proposed towards a final better decellularization quality seeking for the best balance between a “gentle” decellularization, able to maintain an adequate composition of the micro-environmental condition of the ECM, and an ineffective decellularization, leading to cellular and antigenic remnants within the ECM. In 2017, Struecker et al. [64] proposed a perfusion-based, mono-detergent liver decellularization. Interestingly, this group tested this technology under oscillating pressure conditions in which rat livers were harvested and decellularized in a specific device composed of four chambers connected to each other and to a pressure distributor, which mimics oscillating intra-abdominal pressure conditions during respiration. Livers were perfused with a 1% Triton X-100 (5 mL/min) detergent via either the portal vein (PV) or the hepatic artery (HA) over a 3-h period. Decellularization either with (+P) or without (–P) oscillating pressure conditions was tested, resulting in four experimental groups: (1) PV/−P; (2) PV/+P; (3) HA/−P; and (4) HA/+P; (*n*  =  6 for all groups). Arterial liver perfused under oscillating pressure conditions showed a more homogeneous decellularization than livers perfused without oscillating pressure. This result also correlated with a smaller quantity of remaining DNA with a major content in terms of glycosaminoglycans. Different detergent-based protocols have been also evaluated. In particular, Ren et al. [65] tested and compared the cellular removal efficacy of two different protocols. Both were based on a portal vein peristaltic perfusion with the inferior vena cava used as a fluid outlet. The first protocol was based on the use of 1% SDS, whereas the second one exploited a solution of 1% Triton X-100 with 0.05% sodium hydroxide. Decellularization conditions were similar, at 37 °C with 2 h of perfusion and a perfusion rate of 5 mL/min for a total of 600 mL for each sample. The effects on collagen, elastin, glycosaminoglycan (GAG), and hepatocyte growth factor (HGF) content and the influence on the function of hepatocytes cultured in scaffolds were examined and compared. The authors showed that the two decellularization methods successfully removed cells from native liver tissues without leaving any cell nuclei. At the same time, the effects on the quality of liver ECM were different. Specifically, the SDS solution was capable of removing most of the collagen, whereas around 20% elastin, 10% GAGs, and 20% HGF were preserved. In contrast, with Triton X-100-based decellularization, not only most of the collagen, but also 60% elastin, 50% GAGs, and 60% HGF were preserved. In order to test any fallout during the scaffold repopulation, the authors seeded a liver scaffold with a total number of 1.0–2.09 × 10^8^ hepatocytes through the portal inlet without causing significant detectable differences in the engraftment efficiency between the SDS and Triton X-100 treatments (89.7% ± 5.1% and 90.6% ± 5.7%, respectively; *p* = 0.76). In contrast, with respect to liver-specific functions, including albumin secretion, urea synthesis, ammonia elimination, and mRNA expression levels of drug metabolism enzymes, Triton X-100 derived scaffolds reseeded with hepatocytes were superior to SDS scaffolds. They concluded that liver ECM scaffolds constructed by perfusion of Triton X-100 could provide a more effective and ideal scaffold for tissue engineering and RM approaches.

#### 4.1.2. Large Animal Model

In the context of clinical translation, one of the most important issues to overcome is the difficulty of obtaining a clinically relevant sized hepatic scaffold to repopulate. As described by Mazza et al. in 2018, the use of large volumes of bioengineered tissues or organs presents different and major hurdles [66]. Large-volume tissues or organs require an appropriate cellular source population, and consequently, a route of administration that guarantees sufficient oxygen and nutrient supply (more complicated to achieve in a large-volume scaffold). One of the first successful report of porcine decellularized liver scaffold was proposed in 2013 by Mirmalek-Sani et al. [67]. The group adopted a chemical dual-detergent based decellularization, which was previously used for a small-animal model, to decellularize livers from 20–25 kg pigs. Porcine livers were anterograde perfused via the hepatic artery with chilled PBS, Triton X-100 (three cycles with increasing concentrations of 1%, 2%, and 3%) and finally with SDS (0.1%) solutions in saline buffer with a flow rate around 50 mL/min. Histological analysis showed the typical loss of cellularity with a consequent lack of nuclear hematoxylin staining and clearance of cellular cytoplasmic keratins, leaving a collagenous-rich, acellular matrix behind.

Scanning electron microscopy (SEM) verified that an intact liver capsule, which is a porous acellular lattice structure with intact vessels and a striated basement membrane, was preserved. Also, for cytotoxicity testing, biopsy examples of acellular scaffolds were statically seeded with hepatoblastoma (HepG2) cells and cultured for as long as 21 days. At different time-points (days 7 and 21) cells did not reveal apoptotic markers. Cells were discovered to be connected to the matrix surfaces with negligible penetration into the liver matrix scaffold. Furthermore, “naked” liver scaffolds were subcutaneously implanted into rodents in order to explore scaffold immunogenicity with no adverse host reaction in the surrounding matrices. This research showed that with protocols developed for rat livers, effective decellularization of the porcine liver could be accomplished and yield non-immunogenic scaffolds for future hepatic bioengineering research.

Continuing searching prospective clinical applications, Yagi et al. [68] verified that a dual-detergent protocol (SDS and Triton X-100) could be used to acquire a porcine liver acellular scaffold that preserved ultrastructural extracellular matrix elements, functional features of the native microvasculature, and the bile drainage network in addition to important growth factors necessary for angiogenesis and liver regeneration (such as HGF, bFGF, VEGF, insulin-like growth factor). Interestingly, the group repopulated the scaffold using an intra-portal multistep infusion with 1 × 10^9^ hepatocytes. The liver scaffold was decontaminated by ultraviolet (UV) irradiation before perfusion. Culture experiments were performed, and the repopulated scaffold finally moved to a custom organ culture chamber composed of a peristaltic pump, a bubble trap, and an oxygenator. The system was then placed at 37 °C in a temperature-controlled incubator. The oxygenator was connected to a mixture of atmospheric gas. After cellular infusion, the graft was continuously infused with continuous oxygenation through the portal vein at 4 mL/min, resulting in an inflow of partial oxygen tension of about 300 mmHg. The medium was changed daily. Twenty hours after infusion, more than half of the cells attached to the decellularized liver scaffold (attached cells: 74 ± 13 percent of the infused cells) the remaining cells were found in the portal vein but progressed into the parenchymal space on day 4. After four and seven days of culture-perfusion, albumin staining showed that hepatocytes engrafted the nearby the larger vessels and repopulated the surrounding parenchymal area.

After four days of perfusion, the amount of immunostained albumin of engrafted hepatocytes was similar to that in normal livers; however, the expression in the perfusion culture dropped significantly after seven days. These findings assessed the way in which porcine livers can be successfully delivered on a large-scale liver bioscaffold and provide significant insights into the development of a bioengineered liver of human size. More recently, in 2015, Struecker et al. presented a decellularization technology based on a pressure control protocol [69]. The group suggested an accelerated protocol (for a total 7-h perfusion time) via an efficient hepatic artery (120 mmHg) and portal vein (60 mmHg) perfusion for human-scale liver decellularization by pressure-controlled perfusion with 1% Triton X-100 and 1% SDS. The perfusion device enabled the generation of specific pressure conditions mimicking intra-abdominal conditions during respiration, thus optimizing micro-perfusion in the liver and homogeneity of the entire process of decellularization. Uncommonly, two years later, the same group presented a version designed for a small animal model [64].

In their report, Ko et al. [70] concentrated on the method of reendothelializing livers obtained after a Triton X-100 and ammonium hydroxide perfusion-based decellularization. Instead of relying on endothelial cells to passively attach to the native matrix, the investigators actively facilitated their attachment by treating scaffolds with 1-ethyl-3-(3-dimethylaminopropyl)-carbodiimide/*N*-hydroxysuccinimide (EDC/NHS) and by perfusing anti-CD31 antibodies through the vasculature. With these infused antibodies, both static and perfusion culture methods were used to infuse endothelial cells. Not surprisingly, compared to acellular scaffolds, scaffolds reendothelialized using this technique show enhanced patency and resistance to platelet adherence. Therefore, reendothelialized livers implanted in a heterotopic porcine model could withstand 24 h of physiological flow. In addition, for the first time, this study proved that a large-scale decellularized/re-endothelialized liver scaffold can be successfully implanted in a large animal recipient while preserving intra-hepatic blood flow. While it will take a much longer time until these systems are ready for therapeutic use, this article shows the way in which engineering strategies can be used to deliver new cells within decellularized organs to specific regions. The extensive exploitation of pig livers is associated with both their broad accessibility, as well for the dimensions within a range consistent with the size of human liver. For these reasons, several studies organizations have investigated their application for liver bioengineering [71,72,73].

#### 4.1.3. Human Tissue

Even if the use of xenogeneic livers, derived from different species, is largely under consideration and deeply discussed as a template for clinical application, major concerns have been raised based on 3D architectural differences, on biocompatibility, and on immunogenicity. In particular, the difference in vascular structure between human liver and liver collecting from animal species could lead to hemodynamic consequences incompatible with the preservation of the transplanted engineered liver tissue. Indeed, the ideal biomaterial should be derived from human liver. Following this concept, in 2015 Mazza et al. [56] applied decellularization to a whole human liver and successfully obtained a human whole-liver acellular ECM-based scaffold for the first time. The decellularization protocol consisted of a perfusion regime based on repeated perfusion cycles of distilled water (dH_2_O), para acetic acid (PAA), and ethanol (EtOH), preceded by a single cycle of freezing/thawing. Histological staining (H&E, Sirious Red, and Elastin Von Gieson) demonstrated the complete cellular removal with the preservation of type I, III, and IV collagen, fibronectin, and elastin in the decellularized sample. Ultrastructural characterization by SEM also confirmed the preservation of the 3D microanatomy of the portal tract with a surrounding honeycomb-like pattern. The authors explored the interspecies biocompatibility of the scaffold through the subcutaneous implantation of 125 mm^3^ cubic ECM scaffold fragments in immunocompetent C57BL/6J mice and finally evaluated the fragments at 7 and 21 days after implantation. Seven days after implantation, polymorphonuclear cells and lymphocytes were noted, suggesting a mild inflammatory response. Inflammatory cells were mostly observed just in the tissue around the implants. By contrast, little or no inflammatory infiltrate was noted around the implants at 21 days after implantation. The omental implantation of the scaffold fragments also confirmed these results, indicating progressive infiltration of host cells and arteriolar neovascularization. While further studies are needed, it must be stressed that biocompatibility is a crucial issue which needs to be clarified when proposing biotechnologies that might lead to advanced therapeutic medicinal products (ATMP) [74]. Human-liver cubic scaffolds have been also statically seeded by the hepatic stellate cell line, LX2, the human hepatocellular carcinoma cell line, HepG2, and the liver adenocarcinoma cell line, SK-Hep1 (for a total amount of 2 × 10^6^ cells). Seeded scaffolds were kept at 37 °C for 2 h in a humidified atmosphere with 5% CO_2_ to facilitate cell attachment, followed by the addition of a complete culture medium up to 21 days after seeding. H&E and Ki67 staining showed that all cell types were able to repopulate liver scaffolds while still proliferating at 21 days. Cellular well-being was also confirmed by a significant increase in the total cell count between 7 and 14–21 days in human liver scaffolds repopulated with LX2, HepG2, and Sk-Hep1 cell lines. Again, the same team suggested a new decellularization protocol based on implementation of elevated shear stress, which would speed up cellular removal from human liver tissue [75]. In this case, decellularization was focused on samples of liver cubes. After the first perfusion with 1% PBS in order to eliminate blood, livers were frozen at −80 °C for a minimum of 24 h to assist with the destruction of the cellular membrane. Afterwards, the livers were thawed at 4 °C overnight and cut into 125-mm^3^ cubes and then underwent a cycle of freezing/thawing. Once thawed, the cubes were transferred into 2-mL safe-lock tubes and an increasing g-force intensity (45*g*) was applied with an orbital shaker. With this setup, 3 h of shaking were enough to obtain acellular liver scaffold cubes (compared to 36 h in the previous perfusion protocol). Histological analyses confirmed the elimination of both nuclear and cellular materials with concurrent conservation of collagen and elastin after decellularization. SEM analyses showed the preservation of the portal tract, collagen fibrils, and hepatocyte pockets. The capacity of ECM scaffolds to encourage neo-angiogenesis was demonstrated through the use of the chicken chorioallantoic membrane (CAM) assay in several studies via the formation of new blood vessels in spoked-wheel structures close to the scaffold.

Later on, Verstegen et al. [38] continued to exploit the use of marginal human livers, considered medically inappropriate for transplantation due to poor quality, as a prospective source of bioengineered hepatic scaffolding.

Aiming to perform whole liver decellularization in a clinical series, they proposed the use of a dual perfusion through the portal vein and hepatic artery via a custom-made controlled machine and were able to produce a mild nondestructive decellularization protocol. In 11 discarded human whole liver grafts, this protocol was demonstrated to be effective for generating constructs that reliably maintain hepatic architecture and ECM components using machine perfusion while completely removing cellular DNA and RNA.

While excellent results have been documented in the literature about the use of human liver as the basis for hepatic bioengineering, much work still needs to be done and potential advances for in-vitro recellularization may be achieved by innovative and dedicated bioreactors to better replicate the physiological microenvironment of native liver.

## 5. Recellularization Technology

In order to create an organ suitable for transplantation, the decellularized scaffolds should be repopulated with suitable functional cells capable of performing all of the organ-specific tasks. Specifically, this seems to be difficult for the liver due to the multitude of different rules that it fulfils, including, above all, metabolic processes and albumin and cholesterol production. Scaffold recellularization is a crucial step in OBE in which cell types, sources, numbers, and seeding methods need to be seriously evaluated [76]. Recellularization of whole liver is currently progressing. While recapitulation of sinusoids is ongoing with hepatocytes (which accounts for about two-thirds of the total quantity of the liver) [77], biliary tree regeneration on liver ECM has demonstrated more challenges. After undergoing damage, hepatocytes are confirmed to be allowed to physiologically regenerate liver processes and functionality [78]. As a result, primary hepatocytes may be the first-line option for liver ECM scaffold recellularization. However, due to their partial proliferative capacity, maintaining and expanding in vitro hepatocytes appear to be troublesome as they are only viable for few weeks [79]. Furthermore, once coated, these cells slowly lose the typical morphology in addition to liver-specific features, including protein synthesis, carbohydrate metabolism, and cytochrome P450 activity in a process called de-differentiation. Finally, in order to prevent rejection, and with the objective of producing a fully immunological human compatible organ, hepatocytes must be freshly isolated from the final recipient (the patient). In fact, it has been described that after hepatocyte transplantation, the recipient may undergo inflammatory reactions and immune rejection in the first 24 h [80]. In recent years, several attempts have been produced to discover another cellular source of scaffold repopulation. Stem cells are encouraging substitutes of primary hepatocyte because of their self-renewal capacities and their capability of giving rise to different type of cells [81]. In the literature, several protocols using stem cells have been described for reproducing functional hepatocytes. These methods are based on addition to the culture medium of specific soluble factors, such as growth and transcriptional factors and cytokines [82]. Moreover, some results have indicated that differentiation of stem cells into mature hepatocyte is more efficient on a 3D scaffold compared with 2D conditions [83,84,85]. Despite several efforts, until now, it has been observed that the different types of stem cells used for recellularization can give an unlimited number of hepatocyte-like cells (HLCs) but with incomplete functions [86].

### 5.1. Cell Sources

HLCs used in liver bioengineering can be generated from several sources:
Embryonic stem cells (ESCs)Hepatic progenitor cells (HPCs)Fetal stem cellsMesenchymal stem cells (MSCs)Induced pluripotent stem cells (iPSCs).

#### 5.1.1. Embryonic Stem Cells

ESCs are pluripotent stem cells derived from the inner cell mass of blastocysts [87]. Specific cell type differentiation begins from the creation of three germ layers: (1) endoderm; (2) mesoderm; and (3) ectoderm. ESCs could be cultured indefinitely in an undifferentiated state and they can differentiate in vitro into hepatoblasts in the presence of specific stimuli and gain expression of characteristic liver cellular markers [88,89,90]. However, there are ethical limitations for their use, and they are characterized by the loss of epigenetic modifications, which could cause development of teratomas [91].

#### 5.1.2. Hepatic Progenitor Cells

HPCs are classified as adult stem cells that are partially hepatic committed. HPCs present a greater regenerative capacity than adult hepatocytes and have a physiological role in liver tissue repair after damage. They naturally show bipotential differentiation capabilities in both hepatocyte and cholangiocytes, two main epithelial liver cell types [92]. HPCs can be isolated and expanded from discarded livers. They are difficult to isolate due to the absence of specific markers. Khuu et al. [93] described that after in vitro differentiation, HPCs expressed albumin, alpha fetoprotein (AFP), and cytokeratin 18 (CK18), supporting their hepatic commitment. Wang et al. [57] demonstrated that human hepatic stem cells seeded onto liver specific biomatrix scaffolds in the presence of specific stimuli lost stem cell markers and differentiated into mature liver parenchymal cells.

#### 5.1.3. Fetal Stem Cells

Fetal stem cells are multipotent cells that are able to differentiate into hepatocytes and cholangiocytes under specific stimuli [94]. They can be isolated from fetal tissues in addition to fetal blood and basement membranes. They show higher clonogenic and lower immunogenic potential in vitro than adult stem cells [95]. Moreover, their differentiation potential seems to be greater than their adult counterparts. Several studies have shown the capability of fetal stem cells to give rise to HLCs. Baptista et al. [96] reported that hepatoblasts seeded onto bioscaffolds differentiated into the biliary and hepatogenic lineage. Zhang et al. [90] obtained HLCs from human fetal stem cells, which gained in vitro functional activity, such as albumin production, glycogen storage, and CYP450 activity. However, the limited cell number that could be obtained and the ethical issues that arise from the use of fetal stem cells represent great limits for their applications [95].

#### 5.1.4. Mesenchymal Stem Cells (MSCs)

Mesenchymal stem cells (MSCs) have recently been reported as a suitable source of cells for liver bioengineering. MSCs are multipotent stem cells that generate numerous mesodermal cell types. MSCs can be isolated from various sources, including bone marrow, umbilical cord blood, adipose tissue, liver, spleen, trabecular bone, and joint cartilage [97]. The benefits of using MSCs in tissue RM include simple isolation, high proliferation capabilities, and ultimately the prospective use of autologous cells. It has been noted that human MSCs can differentiate into HLCs under appropriate in vitro culture conditions [98,99]. In addition, MSCs can induce endogenous parenchymal cell regeneration and increase fibrous matrix deterioration [100]. Moreover, there are no ethical or tumorigenic concerns. Recently, Li et al. [101] investigated the influence of decellularized liver scaffold on hepatic differentiation of derived umbilical cord blood MSCs. Hepatic gene marker expression, such as that of albumin, CK18, hepatocyte nuclear factor 4 (HNF4), and two isoforms of CYP450 (CYP1a2 and 3a4) was upregulated after 25 days of the induction protocol, while stem cell-specific genes such as octamer transcription binding factor (Oct)-4 and sex-determining region box (Sox)2 were downregulated. Moreover, MSC-derived HLCs gained liver specific functions, such as albumin secretion, glycogen storage, and ammonia to urea conversion.

#### 5.1.5. Induced Pluripotent Stem Cells

iPSCs, derived from somatic cells that are reprogrammed to the pluripotent state, are being widely explored as alternatives to primary human hepatocytes for their capacity to differentiate in vitro into HLCs in the presence of specific stimuli [102]. iPSCs present the same level of pluripotency as ESCs and constitute an unlimited cell source that gives rise to both parenchymal and supportive cells [103]. Furthermore, iPSC technology in OBE has the advantage of creating patient-specific cell-therapy [104]. The use of iPSCs have no ethical problems and do not induce host immune rejection [105]. At first, Yamanaka et al. [106] demonstrated that both mouse embryonic and adult fibroblasts could be genetically reprogrammed to create IPSCs by the retroviral transduction of four reprogramming factors (Oct4, Sox2, Krueppel-like factor (Klf)4), and the regulator (c-Myc, genes). Currently, there are different protocols for creating iPSCs based on non-integrating genomic modifications, protein introductions, and use of chemical agents [107,108]. Recently, Jaramillo et al. [53] evaluated the effects of decellularized human liver extracellular matrix for increasing the efficacy of a differentiation protocol towards HLCs. Under these culture conditions, functional hepatic markers were upregulated, and hepatic transcription and nuclear factor expressions were similar to those of primary human adult hepatocytes. However, even though these cells may represent an ideal source for repopulation, iPSC-derived HLCs are still not functionally equivalent to primary hepatocytes. In addition, the tumorigenic potential of iPSCs must be evaluated.

In 2019, Kehtari et al. [109] proposed a protocol for obtaining hepatic differentiation of hiPSCs seeded on decellularized Wharton jelly (DWJ) scaffolds as an alternative scaffold for circumventing donor shortage. Cell volume increased significantly after differentiation, showing the typical morphological characteristics of hepatocytes. In addition, the expressions of human liver-specific genes, such as albumin, CK19, the regulatory protein TAT, and CYP7A1, increased considerably when cultured on DWJ scaffolds (compared to the corresponding 2D conditions). In addition, albumin secretion and urea synthesis were evaluated for testing the functional and metabolic activities of hepatocyte-derived iPSCs.

Even though HLCs in DWJ scaffolds exhibited more abundant and stable metabolic activities than those cultured on culture plates, iPSCs still presented lower functional capabilities when compared to primary hepatocytes.

### 5.2. Cell Seeding Strategy

Recellularization efficiency is also influenced by seeding method and cell number. Since hepatocytes are able to regenerate, a number of cells between 1% and 10% of native liver mass is enough for organ reconstitution [110]. In addition, the need arises to maintain different cell type proportions in order to guarantee a physiologically functional organ. The scaffold may be repopulated by different cell types following different seeding methods. The most commonly used methods for recellularization include direct parenchymal injection, multistep infusions, and continuous perfusion. Soto-Gutierrez et al. [111] tested these methods and determined that efficiency with multistep infusions was higher than with direct parenchymal injection or continuous perfusion. In another study, Uygun et al. [52] evaluated the efficiency of mouse hepatocyte reseeding onto decellularized rat livers using direct parenchymal injection, continuous perfusion, or multistep infusions. After extensive evaluation of the integrity, attachment, function, and distribution of engrafted cells, it was found that the multistep infusions technique offered the most suitable results.

Recently, bioreactors have been receiving increasing interest as a strategical tool for preserving recellularized whole-liver scaffolds [112]. A bioreactor is a type of organ-perfusion system that provide a continuous supply of nutrients and oxygen while concurrently removing metabolic waste. Ideally, a bioreactor system should be capable of maintaining a full recellularization in a whole-organ scaffold in terms of temperature, perfusate, chemical factors, and mechanical environment. Several factors, such as flow rate and perfusion, should be well defined as they have a major impact on cell or tissue growth.

A critical aspect of recellularization is determined by the loss of the organ endothelial layer due to the decellularization process. In the absence of such cells, coagulation can start when blood is exposed to matrix proteins. For this reason, it is essential to develop strategies that improve the hemocompatibility of the scaffolds and avoid blood clotting in the vascular system of the transplant recipient animal model. Based on these observations, a co-culture system of vascular cell type with MSC in order to obtain a vascularized tissue construct was described [113]. Hussein et al. [114] developed a heparin–gelatin mixture in order to cover vascular surfaces in decellularized porcine livers. After coating the blood vessel, scaffolds were reseeded with endothelial cells (EA.hy926) and subsequently with epithelial cells derived from a HepG2 cells line. Their results demonstrated that a heparin–gelatin gel supported the attachment and migration of endothelial cells. The decellularization and recellularization techniques considered in the present review are summarized in Table 2 and Figure 3.

## 6. New Perspectives

### 6.1. 3D Bio-Printing of Human Hepatic Tissue Using Liver Extracellular Matrix as Bio-Ink

Three-dimensional bio-printing is a combination of 3D printing and tissue engineering through the use of biomaterials, biochemicals, and living cells as “ink”. The goal of this growing field of research is the reproduction of tissues or whole organs for drug development and testing, disease modelling and, most of all, RM. So far, 3D bio-printing technology has been exploited for the development of many in vitro tissues, including skin [124,125], nerve grafts [126], cardiac tissues [127], vascular tissues [128], and bone tissues [129].

The currently in-use bio-ink is a hydrogel solution with cells suspended in it; the challenge is the development of a hydrogel with characteristics as similar as possible to the natural ECM, which varies from tissue to tissue.

Despite the different preparations tested up to now, a perfect combination has not been found. Naturally derived bio-inks (such as agar, agarose, collagen, gelatin, alginate, chitosan, hyaluronic acid, and fibrin/fibrinogen) have low viscosity, which makes them unsuitable for bioprinting. On the other hand, synthetic materials (such as PU, PEG, and PLLA) have tunable mechanical properties and cross-linking capacities, but they are not adequate for cell adhesion, growth, differentiation, survival, and function. Therefore, decellularized ECM has been introduced as an alternative source for bio-ink. The advantages of using animal or human ECM is the maintenance of tissue microenvironment with retained growth factors and cytokines that act as biological cues for cellular activities. Furthermore, the native ECM is capable of inducing tissue repair and avoiding antigenicity-related reactions of the host tissue against the graft [66,130]. Bio-printed tissue with decellularized ECM (dECM) has also shown to have a biodegradation rate equivalent to that of the same in vivo tissue, balanced with the cells’ capability to secrete new ECM. The main problems with dECM are the rheological properties, which are important for cell survival during the extrusion phase and for the preservation of the shape of the printed module. dECM is soft and possesses poor mechanical properties. However, it has been demonstrated that vitamin B2 addiction or ultraviolet (UV)A radiation can induce covalent cross-linking and allow for tuning of the mechanical behavior [131]. In 2017, Lee et al. [132] developed a liver-decellularized ECM for 3D bio-printing. In this study, a porcine liver was decellularized, lyophilized, and ground into powder. The resulting powder was then solubilized by pepsin digestion with 0.5-M acetic acid and centrifuged to remove undissolved particles, which would have eventually blocked the nozzle of the 3D printer. Finally, liver dEMC solution was neutralized in order to obtain a pH favorable for cell encapsulation; human bone marrow-derived stem cells were used. The dECM showed excellent print capabilities without significant cell death during the printing phase. Four liver-specific transcription factors (HNF1A, HNF3B, HNF4A, and HNF6) were analyzed and compared to those expressed by the same stem cells encapsulated in a collagen bio-in, showing an enhanced stem cell differentiation in the dECM [133].

### 6.2. Three-Dimensional Organoid Culture Environment: The Concept of Stem Cell-Driven Tissue Engineering

A new strategy for regenerating a functional and transplantable liver graft is based on liver organoids. Organoids are defined as an organized 3D structure derived from different stem cells in which cells spontaneously self-organize into multiple functional cell types or progenitors, acquiring some characteristic functions of native tissue [96,134,135,136]. In this way, organoids mimic the in vivo structure and complexity of an organ.

Organoids could be obtained from fresh or frozen patient biopsies via the use of low-invasion techniques [137]. Several characteristics make adult human liver organoids suitable for cell therapy approaches: (1) they have an extensive degree of clonogenic potential; (2) cells maintain the capability of differentiating into hepatocytes or cholangiocytes under organoid culture conditions; (3) proliferating cells could be maintained for months without genetic transformation (genetic stability); and (4) a significant number of cells can be obtained from a small number of starting cells, such as a sample from a liver biopsy.

However, there are disadvantages, such as difficulties in standardizing cell cultures and ethical issues for ESCs-derived organoids. Moreover, the lack of vasculature system, stromal cells, and immature cells represents a problem.

Organoids can be produced by different strategies [137]: differentiating cells on-bed of feeder cells, self-assembling on ECM-coated surfaces, using mechanically assisted culture for primer tissue differentiation, forming embryoid bodies over a hanging drop culture system, and using floating culture of embryoid body-like aggregates on low-adhesion plates in serum free-condition.

In their report, Huch et al. [136] observed that under specific organoid culture conditions, p-leucine-rich repeat-containing G protein-coupled receptor 5 (Lgr5)-positive cells, under the influence of Wnt signaling, spontaneously self-organized into specific structures called cyst-like organoids. These structures mimic the functionality and cellular composition of liver tissue and can be maintained in vitro for several months to a few years. Cells in the 3D liver organoid in a specific medium could differentiate into hepatocytes or cholangiocytes. Upon hepatocyte differentiation in organoids, cells gain hepatic morphology and present an upregulation of typical hepatocyte and ductal markers. Moreover, organoids show some hepatic functions, such as glycogen storage, albumin production, and low-density lipoprotein (LDL) uptake although to a smaller degree than mature hepatocytes.

#### Patient-Derived Organoids for Personalized Applications

Organoid techniques might be useful in different clinical applications such as disease modeling, drug screening, and clinical implantations.

To date, several studies related to drug efficacy and safety have been performed on animal disease models. Recently, the development of 3D liver stem cell cultures organized in organoids provided opened new options in this field [138], allowing the evaluation of some pathological aspects without the use of more expensive and time-consuming animal models. In fact, cells obtained from patients with genetic liver disease also maintained the disease phenotype in culture.

Autologous transplantation of genetically corrected organoids from patients with metabolic disease could be another therapeutic application of a liver organoid. In this regard, Huch et al. [135] demonstrated that organoids derived from patients with alpha1 antitrypsin deficiency (AA1T-D) reproduced the same misfolding and aggregation of hepatocyte alpha1 antitrypsin protein (AA1T) in the organoid structures.

Moreover, Nantasanti et al. [139] obtained gene correction by transferring the copper metabolism domain-containing 1 (COMMD1) gene to COMMD1-deficient organoids obtained from dogs with autosomal recessive COMMD1 deficiency as in Wilson’s disease model.

In particular, liver organoids have been used to study monogenic diseases of epithelial compartments [140]. However, in clinical applications, genetically corrected organisms might cause graft rejection and for this reason, short-term immunosuppressive therapy is recommended despite autologous transplantation in order to avoid immune reactions to non-self-proteins in transduced organisms. Furthermore, genetic instability might increase the risk of tumor formation following transplantation [135].

### 6.3. Application of ECM-Derived Livers as A New Tool for Drug Testing

An ideal model for pharmacological testing should have two characteristics: (1) it must resemble a human being as much as possible and (2) at the same time, it must preserve the integrity of the tested subject. Animal models have been consistently used but due to the considerable interspecies variability, together with the ethical and financial issues, they were not found to be ideal for toxicological and pharmacological drug testing. At the present time, the recognized gold standard for in vitro testing of drugs is primary cultures of human hepatocytes.

In order to allow for optimal phenotypic gene expression and response to drugs, histological and physiological hepatocyte conditions, in addition to their interactions with one another, should be maintained in in vitro studies. In order to recreate these conditions, different configurations of ECM and medium have been tested. However, the reproduction of a natural-like ECM is a challenge due to the large variety of constituents produced by different cell types of the liver. In such cultures, cells tend to lose their morphology and integrity with time; after approximately two weeks, hepatocytes deteriorate due to cytoskeletal alterations, which subsequently cause changes in the signaling pathway and eventually, in phenotypic gene expression [141]. The result is down regulation of drug-metabolizing enzymes, thus reducing the relevance of the testing. In order to overcome these limitations, bioengineered livers could be used in the near future for research purposes [57,142,143]. Liver bioscaffolds have been seeded with both mature hepatocytes and stem cells. Hepatocytes plated on decellularized liver scaffolds have significant advantages compared to primary hepatocyte cultures. The natural ECM not only promotes the 3D disposition and orientation of cells but also allows the cells to interact with the matrix-bound cytokines and growth factors, which are preserved after decellularization. In these conditions, hepatocytes have been found to attach faster to the bio-matrix (few minutes versus hours on type I collagen) and remain fully functional and morphologically stable longer (eight versus two weeks on collagen type I) [57].

The preserved 3D architecture and native composition of the decellularized liver scaffold are able to drive differentiation of seeded stem cells towards adult liver fates. Stem cells have self-renewable ability, high proliferative potential, and the possibility to differentiate into multiple cell lineages; these features give to this in vitro liver model a constant availability of cells, thus allowing longevity of the culture and presence of hepatocytes together with biliary epithelial cells [144]. A recent study investigated the thus-generated liver by exposing the liver to six drugs that are well-known for targeting specific CYP enzymes and found that these drugs provided enhanced activities of metabolic enzymes compared to the 2D culture conditions [145]. In summary, it was concluded that liver scaffolds “better represent the natural in-vivo environment in an ex-vivo system” and offer tremendous opportunities for drug development and testing [116]. Furthermore, the use of patient-derived stem cells allows for the possibility of testing patient-specific responses to drugs.

## 7. Conclusions

The main goal of liver bioengineering remains to provide new functional organs for clinical translation in order to overcome the shortage of organ donors. Currently, to achieve this purpose several challenges need to be met. Within this journey towards bioengineered organs, there are three fronts in the preclinical setting on which to work: (1) scaffolding; (2) recellularization; and (3) cell signaling. The investigations concerning the ECM reveal that a state of dynamic reciprocity between cells and ECM exists. It will therefore be necessary to discover the complex and delicate dynamic equilibrium between the cells and ECM, allowing the generation of new organs. In particular, a more detailed decellularization process, an optimization of organ-specific recellularization techniques, a better cell differentiation capacity, and a more exhaustive understanding of the interaction between cells and ECM will play an essential role in the progression of this research field. Moreover, scaffold-based organs and 3D printed organoids provide an alternative method for studying liver disease and are suitable for personalized medicine. Currently, the possibility of using hepatic organoids in cell therapy is encouraging but requires further verification in clinical settings.

Then, the other great challenge will be the translation to the clinical setting that will hinge on how effectively we understand “the bench phase”.

## Figures and Tables

**Figure 1 biomolecules-09-00813-f001:**
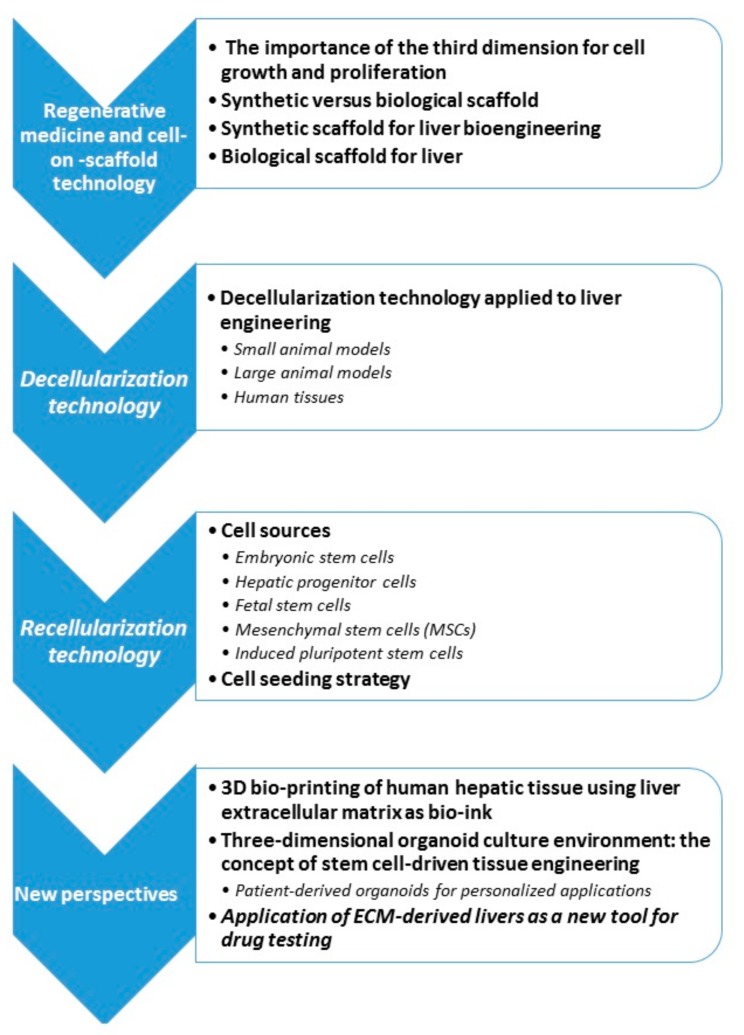
Review organization. ECM: extracellular matrix.

**Figure 2 biomolecules-09-00813-f002:**
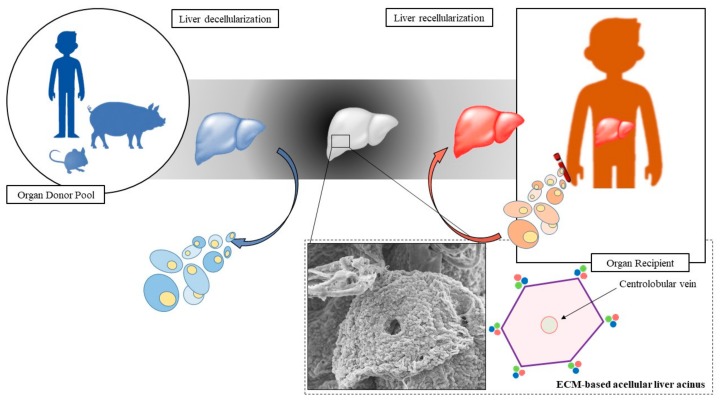
Steps for liver bioengineering. Organs may be obtained from donor pool, decellularized, and then recellularized for liver transplantation.

**Figure 3 biomolecules-09-00813-f003:**
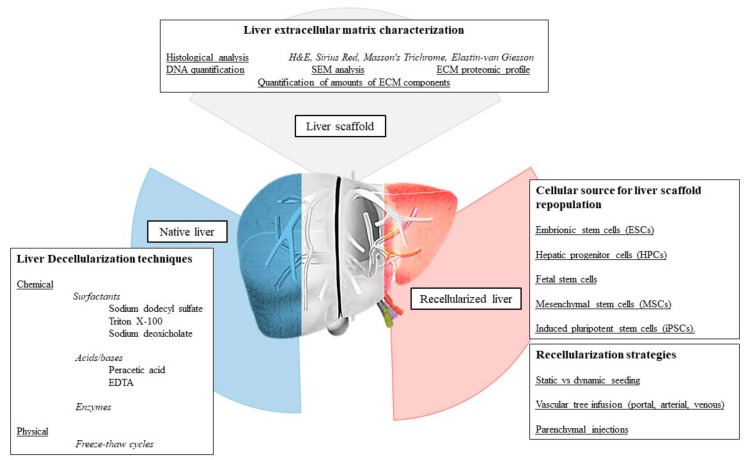
The decellularization and recellularization process. Methods for liver extracellular matrix evaluation after decellularization are reported together with the strategies for decellularization and recellularization. Cellular sources for liver scaffold repopulation are presented. H&E: hematoxylin and eosin.

**Table 1 biomolecules-09-00813-t001:** Advantages and disadvantages of synthetic vs. biological scaffolds.

	Synthetic Scaffolds	Biological Scaffolds
Advantages	-sterilizable -economical -easy to synthesize -do not require organ donors -no pathogenicity	-sterilizable -native organ structure -active molecules useful for cell growth (peptides and ECM-specific proteins) -not immunogenic
Disadvantages	-cell apoptosis in the absence of a vasculature system -difficulty to reproduce organ complexity -biocompatibility	-organ donors -standardization of optimal decellularization conditions

**Table 2 biomolecules-09-00813-t002:** Decellularization and recellularization techniques in different experimental models.

Scaffold Origin	Decellularization Technique and Agents		Cell Source	Recellularization Technique and Culture Time (days)	Ref.
Rat	PV-p	SDS	Rat hepatocytes	PV-i (7)	[52]
Rat	PV-p	1%, 0.5%, 0.25% SDS + 1% Triton X-100	Rat hepatocytes	PV-i (0.25)	[115]
Rat	IVC-p	3% Triton X-100/0.5% EGTA	Mouse hepatocytes	DI/PV-p (7)	[111]
Ferret	PV-p	1% Triton X-100/0.1% NH_4_OH	h-fetal liver cells + h-UVEC	PV-i (7)	[96]
Rat	PV-p	1% Triton X-100 + 0.05% NaOH vs. 1% SDS	Rat hepatocytes	PV-i (7)	[116]
Pig	PV-p	0.25%, 0.5% SDS	h-fetal stellate cells + h-fetal hepatocytes	PV-i (13)	[71]
Mouse	PV-p	1% SDS + Triton X-100	h-iPCS	PV-i (14)	[59]
Pig	PV-p	0.01%, 0.1%, 1% SDS + 1% Triton X-100	Porcine hepatocytes	PV-i (28)	[68]
Rat	SVC-p	Trypsin, Triton X-100 + EGTA	Rat hepatocytes + rat BM-MSCs	PV-i (6)	[117]
Mouse	PV-p	1% Triton X-100 + 0.1% NH_4_OH	Mouse BM-MSCs	PV-i (28)	[118]
Rat	PV-p	Triton X-100 + 0.1% SDS	h-liver stem cells	PV, IVC, SVC + CBD-i (21)	[62]
Pig	PV-p	1% Triton X-100/0.1% NH_4_OH	Mouse vascular endothelial cells	PV-i (3)	[70]
Human	IVC-p	3% Triton X-100 + 1% SDS	h-hepatic stellate cells/HepG2/Sk-hep-1	DI (21)	[56]
Rat	PV-p	0.01%,0.1%, 0.2% SDS + 0.1% Triton X-100	Adult rat hepatocytes	DI (5)	[119]
Rat	PV-p	1% Triton X-100 + 0.1% NH_4_OH	h-iPSCs hepatocytes	DI (14)	[63]
Rat	PV-p	1% Triton X-100/0.1% NH_4_OH	Rat liver cell line + h-endothelial cell line	PV-i + DI (7)	[120]
Pig	PV-p	0.1% SDS	Porcine iPSC-heps	PV-i (5)	[121]
Pig	PV-p	0.1% SDS	Hep-G2 + h-endothelial cell line	PV-i, PV-i + HA-i (10)	[114]
Rat	PV-p	0.02% Trypsin/0.05% EGTA + 1% Triton X-100/0.05% EGTA	Mouse fetal hepatocytes	CBD-i (7)	[122]
Mouse	PV-p	1% SDS + 1% Triton X-100	Mouse hepatocytes	PV-i (7)	[123]
Mouse	PV-p	4% SDC +2000 kU DNAse-I	h-ESCS and iPSCs	DI (13)	[60]
Human	a	SDS, Triton X-100, SDC, DNAse	h-hepatic stellate cells/HepG2/hepatocytes	SS/p (14)	[75]
Human	PV-p + HA-p	4% Triton X-100/1% NH_4_OH	h-UVECs	SS (5)	[38]
Pig	PV-p	1% Triton X-100/0.1% NH_4_OH	Pig UVECs/MSCs/hepatoblasts	PV-i + HA-i (21)	[55]

PV-p, portal vein perfusion; SDS, sodium dodecyl sulphate; IVC-p, inferior vena cava perfusion; EGTA, ethylene glycol tetraacetic acid; NH4OH, ammonium hydroxide; NaOH, sodium hydroxide; iHPCs, immortalized mouse fetal hepatic progenitor cells; HA, hepatic artery; SVC-p, superior vena cava perfusion; h: human; BM, bone marrow; MSCs, mesenchymal stem cells; CBD, common bile duct; HepG2, liver hepatocellular cells; Sk-Hep-1, human hepatic adenocarcinoma cells, iPSCs, induced pluripotent stem cells; SDC, sodium deoxycholate; ESCs, embryonic stem cells; UVECs, umbilical vein endothelial cells, PV-i portal vein infusion; DI, Direct injection CBD-i, common bile duct infusion; a, agitation; HA-i, hepatic artery infusion; SS, static seeding.

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
