# Peer review of "A Hepatic Scaffold from Decellularized Liver Tissue: Food for Thought"

_biomolecules, 2019, doi:10.3390/biom9120813_

Round 1

Reviewer 1 Report

The review of Croce et al. offers to readers of Biomolecules a detailed description of recent advances in cell-on-scaffold technology for liver bioengineering, covering the most important aspects of this subject such as, the advantages and disadvantages of synthetic and biological scaffolds, the potential of biologic extracellular matrix-derived scaffold and the processes employed for decellularization. The article is clear, concise and easy to read and understand. It also covers the most important aspects regarding the role of 3D for cell growth and proliferation and the advances obtained with small animal models and the importance of advanced studies focusing large animal models.  Considering the importance of the recellularization, and supported by the appropriated literature, the authors presented the most important aspects of the recellularization technology, discussing the most important cell sources and the cell seeding strategies.  In the section “Bioengineered ECM-derived livers as a new tool for drug testing” the authors discussed how the natural ECM and its 3D disposition favor the interaction of cells with the matrix and drive differentiation of seeded stem cells. Overall, the review is a good contribution, since it offers to readers a concise and clear view of the state-of-the-art of hepatic scaffold obtained from decellularized liver tissue.

Minor points

Page 4, line 157: The phrase “ Even the technique to produce the scaffold is …” seems incomplete.

Page 4, line 174: Mechanical properties instead of  “mechanical proprieties”

Table I is very interesting, however, it should be improved because its quality is not good, font size does not allow reading.

Page 7,  line 224 :  then instead of than

Page 10 , line 422  : “in a process method” ????

Author Response

Geneva

November the 23rd, 2019

To the attention of the Editor in Chief of BIOMOLECULES

Dear Editor,

Enclosed, please find our copy of the revised manuscript manuscript entitled

Hepatic scaffold from decellularized liver tissue: food for thoughts

Our group submitted this review article to your journal, it was rapidly peer-reviewed highlighting the following remarks and criticisms that we completely addressed. For this, we thank the Reviewers for their valuable and useful evaluation and remarks.

Amendments and changes were made and we really hope the revised version will now be suitable for publication in Biomolecules.

Our point-by-point reply to the reviewers in enclosed in this letter.

We thank you for your continuing consideration and look forward to hearing from you soon.

Thank you and best regards,

Andrea Peloso

-----------------------------------------------------------------------------------------

Reviewer 1

We sincerely thank the Reviewer 1 for the careful and extensive revision.

Page 4 line 157 - the sentence has been rewritten (Page 5 line 181-183) Page 4 line 174 - "proprieties" has been corrected with "properties" (page 6 line 203) Table 1 has been re-edited using abbreviations and higher font size Page 7 line 224 - "than" has been switched into "then" (Page 7 line 252) Page 10 line 422 - the word "method" has been delated (Page 11 line 451)

Reviewer 2 Report

This article is a review on past and current strategies to construct liver scaffolds, especially from decellularized liver tissues. The authors present the different methods for (a) decellularization and (b) recellularization with different cell types. This is useful for those working in the field. 

Although the review is actually well-documented and its state-of-the-art is well updated, it is relatively difficult to follow as presented, mainly because it has no particular "aim" presented. At least, it was not clear to me after reading it several times what the aim actually was, other than enlisting the techniques one by one. There is no real discussion, there is almost no real comparison between the different techniques, that a reader of such a review paper would hope to get. The contribution looks too light in my opinion.

I would also recommend the particular following points to be considered, as general comments:

- The insight of the paper is actually very poor. It is difficult to understand the objective of the paper; whether it is about looking for alternatives to transplant (such as functional liver units for substitution), drug-testing platforms (even for precise medicine for instance) or only about presenting the different existing approaches.
In any case, I believe it is very important to state the goal of the review from the beginning and focus on responding to this objective in the text. Otherwise it is only a list. "Liver bioengineering" is mentioned several times a potential objective but it is not clearly defined (in my opinion). Line 91, the physiological functions that are required in bioengineered livers are not even mentioned. These could be the specifications (minimum required) that could help select the best options depending on the goal(s)... but they are left to the readers.

- The "methods" in section 2 clearly shows that the authors only searched a few terms to select papers (this seems to be THE method used) but the readers have no clue what they were looking for in these papers... This points towards a potential listing of the techniques only, which is too poor for a review in my opinion (although the list is quite good and seems exhaustive, as much as I can be aware of, being from this field). The contribution of the paper thus seems light.

- I did not understand the organization of the paper (sections) and I also believe this comes from the lack of a clear objective.

- Table 1 (the only table of the manuscript) that is not really discussed and does not provide any information on the advantages and drawbacks of each solution.

- Also, the pictures 1 and 2 are not in the manuscript and it is not clear which is which if they are the ones in the supplementary material.

Some more detailed comments to take into account:
- L44-45: this would require references
- L51-53: this needs to be more discussed, as some consider this an excellent alternative to animal testing and precise medicine solutions
- L53-54: the argument that BAL and liver-on-chip platforms are not good solutions because they are not performing well in the long run is not acceptable as such as no solution is good in the long term. Also, they are not aiming at substituting transplantation either. Bhatia's work is very successful at function substitution in murine models with liver dysfuntion and her results are very impressive and more promising than many other works presented as successful in the review. The authors have to be more careful with their statements and all depends on the objective set in their review (which is why I believe it is important to state them from he beginning). 
- L56-59: please explain clearly the difference seen by the authors between liver-on-chip and OBE (give the definition of each according to them).
- L66-68: as said earlier, it is not clear what is meant exactly by "liver bioengineering"
- L84-89: I believe this should be in the introduction, it's only a suggestion.
- L103-110: this is also very superficial and would need more material, especially regarding good, bad and limiting aspects.
- L112-115: these are generalities, and some impressive works have been made in 2D, see for instance the works of Khetani's and Bhatia's.
- L116: it would be interesting to know why more multidisciplinary expertise is required for 3D. I guess the fabrication and material expertise (physico-chemical) is what is meant...
- L124-127: what are the specific advantages and disadvantages of such solutions? Nothing is discussed here
- L138: talking about biocompatible scaffolds, cellular adhesion, migration and proliferation are mentioned as important cues in general but none of these parameters is ever discussed for liver. What is important here?
- L139: it is not clear what "limitation of synthetic scaffolds" that "may be overcome" with ECM-derived ones actually is...
- Shouldn't Section 6 go before Section 5? The authors compare synthetic and biological and then present synthetic scaffolds for liver...
- L153-154: the phrase is very strange as I failed to see how these properties impact large-scale production.
- the fact that all materials listed L155-156 are "suitable" for liver bioengineering is very much debatable. Please discuss that. 
- almost nothing is said about why hydrogels are used (regarding softness, possibility to conjugate them with proteins) except for the "biological properties" similar to those of ECM (which is something I do not understand, as I am not sure what biological properties gels have).
- L185-187: I did not understand this phrase
- please remove the image of Table 1 and place an actual table as it is very difficult to read (loss of resolution)
- L240-241: "balance between a gentle decellularization." and...?
- styles mistakes: L563-568; L625-635

Author Response

Geneva

November the 23rd, 2019

To the attention of the Editor in Chief of BIOMOLECULES

Dear Editor,

Enclosed, please find our copy of the revised manuscript entitled

Hepatic scaffold from decellularized liver tissue: food for thoughts

Our group submitted this review article to your journal it was rapidly peer-reviewed highlighting the following remarks and criticisms that we completely addressed. For this, we thank the Reviewers for their valuable and useful evaluation and remarks.

Amendments and changes were made and we really hope the revised version will now be suitable for publication in Biomolecules

Our point-by-point reply to the reviewers is enclosed in this letter.

We thank you for your continuing consideration and look forward to hearing from you soon.

Thank you and best regards,

Andrea Peloso

-----------------------------------------------------------------------------------------

We thank Referee 2 for constructive comments that made our work clearer and more comprehensive.

We clarified the objective of the review both in the abstract (Line 24-29) and introduction (line 85-92). We substituted “liver bioengineering “with a more general “organ bioengineering (OBE)” reporting the definition in Page 2 line 56-58; The hepatic physiological functions have been reported (Page 3 line 105-106) As suggested, we tried to reorganize our paper: after the introduction the review has been divided into four main sections: Regenerative medicine and Cell-on-scaffold technology, Decellularization technology, Recellularization technology and new perspectives. We added a figure (Figure 1) in which we reported the different sections. We introduced a new table (Table 1) in which we reported the advantages and disadvantages of synthetic and biological scaffolds. Table 1, now table 2, has been reedited in order to make it clearer and easier to read. References of Figure 2 and Figure 3 are reported in the text (Line 243, 577 respectively) L44-45: references have been added (Page 2 line 47) L51-54: organ on a chip has been presented also as an alternative to animal testing. We modulated the sentence that BAL and organ on a chip are not performing well in the long run. Bhatia’s work has been introduced in the text (Page 2 line 53-66) L56-59: liver on chip has been changed with organ on a chip to make the sentence more generic and to compare it with organ bioengineering. (Page 2 line 53) L66-68: liver bioengineering has been changed with organ bioengineering OBE (Page 2 line 78) L84-89: as suggested, this sentence has been moved in the introduction (Page 1 line 34-39) L103-110: more materials have been added (Page 4 line 125-132) L112-115: Khetani’s and Bhatia’s work has been added L116: the referee is right, for multidisciplinary expertise we meant fabrication and material expertise. We did not underline this in the text. L124-127: we introduced a table (table 1) reporting advantages and disadvantages of synthetic and biological scaffolds. L138: The sentence “a biocompatible scaffold is requested to guarantee cellular adhesion, migration, and proliferation with a negligible immune reaction” has been modified (Page 5 Line167) L139: the sentence has been re-written (Page 5 Line168-169) Section 5 (now 3.2 Synthetic versus biological scaffold) is a general introduction about synthetic and biological scaffold, followed by the sections specific for liver. We left these sections organized as before. L153-156: the phrases have been reformulated (Page 5 line 181-183) The reasons by which hydrogels are used have been specified (Page 5 Line 190-192) and we explained their biological properties. L185-187: the phrase has been reformulated (Page 6 Line 213-216) We removed the image of Table 1 (now Table 2) and to make it easy to read, we constructed a new one with abbreviations. L240-241: the sentence has been completed (Page 7 Line 267-271) L563-568 and L625-635: styles mistakes have been removed.

Round 2

Reviewer 2 Report

The authors of the manuscript have replied very rapidly, yet efficiently to the reviewers comments in general, without losing the style and idendity of the review paper, but responding to all the comments point by point. I am not sure figure 1 was required but it is an interesting input to the text and table 2 is much better in its current form.

I believe the manuscript is much clearer now, better organized and even though the specific goal/discussion is very general, in my opinion it is now acceptable for publication, as there is an apparent message, consistent throughout the different sections. 

I would still recommend to use a larger font size for the text in Figure 3, as it is difficult to read properly.

I now recommend the publication of the manuscript in the journal.